# Antimicrobial Stewardship: Leveraging the “Butterfly Effect” of Hand Hygiene

**DOI:** 10.3390/antibiotics11101348

**Published:** 2022-10-03

**Authors:** Adrian John Brink, Guy Antony Richards

**Affiliations:** 1Division of Medical Microbiology, Faculty of Health Sciences, University of Cape Town, Cape Town 7925, South Africa; 2National Health Laboratory Service, Groote Schuur Hospital, University of Cape Town, Cape Town 7925, South Africa; 3Institute of Infectious Disease and Molecular Medicine, Faculty of Health Sciences, University of Cape Town, Cape Town 7925, South Africa; 4Faculty of Health Sciences, University of the Witwatersrand, 1 Jan Smuts Avenue, Johannesburg 2000, South Africa

**Keywords:** antimicrobial resistance, antimicrobial stewardship, infection prevention control, synergy, butterfly effect

## Abstract

It is vital that there are coordinated, collaborative efforts to address the threat of antimicrobial resistance (AMR) and to prevent and control the spread of hospital-onset infections, particularly those due to multidrug-resistant (MDR) pathogens. The butterfly effect is a concept in which metaphorically speaking, small, seemingly trivial events ultimately cascade into something of far greater consequence, more specifically by having a non-linear impact on very complex systems. In this regard, antimicrobial stewardship programs (ASP), when implemented alongside infection prevention control (IPC) interventions in hospitals, particularly hand hygiene (HH), are significantly more effective in reducing the development and spread of AMR bacteria than implementation of ASP alone. In this perspective, we briefly review the evidence for the combined effect, and call for closer collaboration between institutional IPC and ASP leadership, and for well-functioning IPC programs to ensure the effectiveness of ASP.

## 1. Introduction

The prevalence of infections caused by antimicrobial-resistant (AMR) bacteria is rapidly increasing and this, along with the constantly evolving global epidemiology, represents a major challenge [1]. Recently, a systematic analysis provided the first comprehensive assessment of the global burden of AMR and reaffirmed that it is a leading cause of death around the world, with the highest burdens in low-resource settings [2]. There were an estimated 4·95 million deaths associated with bacterial AMR in 2019, including 1·27 million attributable deaths. The six leading pathogens associated with resistance and mortality were *Escherichia coli*, followed by *Staphylococcus aureus*, *Klebsiella pneumoniae*, *Streptococcus pneumoniae*, *Acinetobacter baumannii*, and *Pseudomonas aeruginosa*. 

In this regard, increasing numbers of Gram-negative bacteria (GNB), such as the carbapenemase-producing Enterobacterales (CPE), carbapenem-resistant *P. aeruginosa* (CRPA), and carbapenem-resistant *A. baumannii* (CRAB), which are resistant to all routinely available antibiotics, are being reported [1,3]. As such, these pathogens are now referred to as difficult-to-treat resistant (DTR) GNB, practically defined as treatment-limiting resistance to all first-line agents, specifically all β-lactams, including carbapenems and β-lactamase inhibitor combinations, and fluoroquinolones [4]. Notably, DTR GNB have been associated with increased mortality compared with phenotypes where at least one first-line agent is active [4].

In response, there have been numerous calls for awareness and for strategies designed to attenuate these poor outcomes and to prevent the further development and spread of AMR. Major strides have been made in recent years, including the World Health Organization (WHO) approval of the Global Action Plan on Antimicrobial Resistance in 2015, and in 2019 AMR was declared one of the top 10 public health threats facing humanity [5]. Despite the momentum gained, the AMR crisis still persists, as it is a formidable and multi-faceted problem that has not been fully addressed in terms of effective antibiotic stewardship programs (ASP) across the “one-health” spectrum [5]. 

Compounding this challenge, is that whatever positive traction had been gained in combatting AMR during recent years, the SARS-CoV-2 pandemic has resulted in significant setbacks that have adversely affected progress [3,6,7]. This has also resulted in a major shift in healthcare resources towards controlling the virus that resulted in decreased funding, staff allocation and surveillance for AMR [6], as well as an increase in inappropriate antibiotic use in hospitalized patients prior to and after the development of bacterial co- and superinfections. 

Whereas SARS-CoV-2 infection prevention and control (IPC) initiatives may have contributed to a reduction in AMR rates in some institutions, paradoxically, despite the adoption of strict COVID-19 IPC protocols, this did not occur in most hospitals and networks globally [3]. Multiple hospital outbreaks of infections due to MDR organisms during the pandemic have been described [8]. The factors that predominantly may have contributed included non-adherence to personal protective equipment (PPE) or hand hygiene protocols and to PPE shortages. Environmental contamination due to staff shortages and patient over-crowding were probably also major contributing factors [8].

Concurrent bacterial and fungal infections have played an important role in hospital outcomes during the COVID-19 pandemic [9]. Recently, Bassetti et al. provided a global overview of the prevalence of *S. aureus* and methicillin-resistant *S. aureus* (MRSA) in this population and confirmed that MRSA was a common causative agent of pneumonia in patients with COVID-19 [10]. In terms of GNB, the evidence suggests that the COVID-19 pandemic had a substantial negative impact on the global epidemiology of pathogenic organisms and on AMR due to and accompanied by an increase in hospital-onset infections (HOI) [3]. This was most evident for carbapenemase-producing *K. pneumoniae* (bloodstream infections), CRPA (ventilator-associated pneumonia), and CRAB (all infections). Significant heterogeneity was however apparent, mostly in the large, system-wide, regional, or national comparative assessments relative to single-centre studies [3]. The impact is, however, likely to be substantial and renewed efforts to limit any further increase in AMR rates is urgently warranted [3]. 

In this regard, Timsit et al. recently summarized the available evidence, emerging options, and unsolved controversies for the sequential optimization of antibiotic therapy in the intensive care unit (ICU), the site where collateral damage is most apparent [11]. This narrative review also provided compelling arguments for the elaboration and implementation of hospital-wide effective interventions to improve patient outcomes by a reduction in antibiotic-related selection pressure to control the dissemination of AMR in healthcare settings. To achieve this, a critical enabler would be the leveraging of synergy between ASP, diagnostic stewardship, and IPC in our hospitals globally, irrespective of resources, which is the focus of this perspective [12]. 

## 2. Leveraging the “Butterfly Effect” of Hand Hygiene

In chaos theory, the butterfly effect describes the often-unintended consequence of small changes in the initial conditions of the state of a deterministic nonlinear system, which subsequently manifests as large differences. Conceptually, this small, seemingly trivial event may ultimately result in a non-linear impact on very complex systems [13]. Mathematician and meteorologist Edward Norton Lorenz noted that the concept of the butterfly effect was derived from the metaphorical example of the potential for a minor perturbation such as a distant butterfly flapping its wings several weeks earlier, to influence the exact time of the formation of a tornado and the path it takes [13]. 

In this regard, a number of reviews, which included publications from both high and low-income settings, have clearly demonstrated that ASPs, when implemented alongside IPC measures, and in particular measures to improve hand hygiene (HH) compliance, are significantly more effective in hospitals than the implementation of an ASP alone [14,15,16,17]. This implies that a small, seemingly insignificant intervention, such as the administration of an alcohol hand-rub during the five evidence-based key moments for HH, could lead to a dramatic difference within a larger context, such as a reduction in AMR bacteria and HOIs.

As an example, Baur et al. reported that studies that co-implemented a HH intervention along with an ASP reduced AMR bacteria by 66% versus 17% in those that did not [14]. The diverse HH measures that were implemented in the studies included in this meta-analysis, varied from education to replacement of soap-based handwashing with alcohol-based hand rubs and the substitution of hand-operated with elbow-operated soap dispensers. Moreover, in a systematic review and meta-analysis focusing on the impact of ASP in Asia, the rate of HOIs was significantly decreased in those studies that included a HH programme [15]. Lee et al. reported a reduction of 48% in the rate of HOIs with HH interventions, whereas ASPs without did not protect against HOIs at all [15]. It is also important to emphasise that the effect of HH was observed not only for infections due to MRSA but also for those due to resistant GNB. 

However, data describing the impact of bespoke IPC and ASP interventions on colonization pressure and resistance rates of specific highly resistant bacteria such as extended-spectrum β-lactamase (ESBL)-producing Enterobacterales, CPE, CRPA, or CRAB, are scarce, and if performed may be more useful to direct and inform specific collaborative ASP and IPC interventions. 

In this regard, a systematic review and network meta-analysis of the relative efficacy of strategies for the prevention of AMR-GNB in the ICU provided some insights [16]. In this analysis, types of interventions were grouped into five categories: standard care (STD) (i.e., hand hygiene and/or contact precautions), source control (SCT) (i.e., daily bathing or showering or whole-body washing with chlorhexidine), environmental cleaning (ENV), decolonization (DCL) (i.e., selective oropharyngeal decontamination, selective digestive decontamination) and ASP. 

Compared with the STD arm, a multifaceted strategy comprising STD, ENV, SCT, and ASP was the most effective intervention to prevent MDR-GNB acquisition in the overall analysis, and for each specific type of bacteria [16]. Notably, the synergy of the various interventions was evident in that the addition of ENV to STD and ASP or SCT to STD and ENV, significantly reduced the acquisition of CRAB by 72% and 52%, respectively. In addition, the findings also elucidated a “dose-response” relationship between the number of intervention components and outcomes [16]. A limitation of this study is that the majority of studies included reported on ESBL-producing Enterobacterales and CRAB and were from high-income and upper-middle income countries. 

In contrast, Rizk et al., utilizing existing resources in a low-income setting, recently demonstrated how a multi-disciplinary approach and combined interventions by both ASP and IPC teams led to an overall decrease in *A. baumannii* resistance and a sustained reduction in CRAB rates in a tertiary ICU in Beirut, Lebanon [18]. This involved the concurrent introduction of a carbapenem-sparing initiative and bespoke IPC interventions such as the screening, monitoring, and tracking of CRAB, as well as a focus on compliance with multimodal measures. Of interest, the substantial impact of this collaborative initiative also exemplified how different organisms respond to differing strategies and equally important, that such joint interventions can be implemented successfully in low- and middle-income countries (LMICs).

A scoping review recently assessed ASP and IPC interventions targeting healthcare-associated *Clostridioides difficile* and carbapenem-resistant *Klebsiella pneumoniae* (CRKP) infections [17]. Notably, interventions that targeted *C. difficile* appeared to focus more on stewardship, while those targeting CRKP focused more on screening, isolation precautions, or environmental disinfection as core strategies. Whilst some studies that incorporated multifaceted interventions, including HH or other IPC programs and ASP interventions concurrently, were shown to reduce HOIs, improve the rational use of antibiotics, and reduce mortality, there was limited evidence as to how the interventions influenced compliance with any of the interventions. In this regard, a commitment, and a call to strengthen and expand qualitative research efforts to improve the impact on compliance by ASP and IPC programmes, and the impact on AMR, is vital [19]. In addition, to improve collaborative outcomes and progress the impact of ASP and IPC concertedly, how to perform conjoint ASP and IPC interventions, is equally of paramount importance. 

## 3. Coordinating and Integrating Infection Prevention Control and Antimicrobial Stewardship

The data presented provide overall support for the integration of IPC practitioners into the ASP teams and the necessity of promoting adherence to IPC measures, including the level of HH compliance, prior to the development of an ASP that mostly relies solely on antibiotic processes and outcome measures. [12,14]. The cardinal role of the IPC expert (and the epidemiologist) in the development, justification, and measurement of the impact of an ASP, has been supported by a position statement in 2012 from the Association for Professionals in Infection Control and Epidemiology (APIC) and the Society for Healthcare Epidemiology of America (SHEA), which were updated and re-affirmed in 2018 [12,20,21]. While the key supporting role of IPC programs in synergistically enhancing ASP strategies, and diagnostic stewardship has gained momentum in high- and upper-middle-income countries, this is not necessarily the case in LMICs, where considerable challenges to implementation persist [22]. Renewed efforts at a high-level to address this disparity is paramount to stem the tide of AMR globally. 

Successful implementation of organizational IPC strategies to support ASP and facilitate AMR mitigation and management, irrespective of resource setting, would require decisive organizational change management and integration of the ancient African philosophy of Ubuntu, meaning: ‘I am what I am because of who we all are’ [23]:Firstly, to ensure a hospital-wide culture that promotes behavioral change through recognition that by reducing AMR bacteria together, patient safety is improved.Secondly, to neutralize and amalgamate the clinical, microbiology, and IPC operational ‘silos’ within institutions.Thirdly, to facilitate system-wide, interdependent, and coordinated interventions

Although the responsibilities of ASP and IPC programs are different, collaboration between these groups is essential to promote optimal outcomes. According to Manning et al, the vital work of IPC and ASP cannot be performed independently, and deliberate strategic relationship-building is essential [12]. Furthermore, multiple enablers to leverage synergy between IPC and ASP should be considered and exploited if we are to further the effectivity of amalgamated IPC and ASP interventions (Table 1). In addition, ideally a de novo combined IPC-ASP bundle, represented by a small set of evidence-based interventions for a defined AMR pathogen (e.g., CRAB, CRPA, CPE, and MRSA), patient segment/population, resource, or care setting should be designed and substantiated such that, when implemented together, will result in significantly better outcomes than if implemented individually.

## 4. Conclusions

Multifaceted IPC interventions have been shown to be instrumental in reducing HOI rates, improving the rational use of antibiotics, increasing HH compliance, and reducing mortality in both developed and developing countries. Therefore, leveraging the butterfly effect of effective HH and IPC programs, can accelerate progress towards preventing the emergence and cross transmission of AMR bacteria, and can verify that an effective IPC program is fundamental to successful ASP in any organization. 

## Figures and Tables

**Table 1 antibiotics-11-01348-t001:** Enablers to leverage synergy between infection prevention control and antibiotic stewardship.

Identifying, defining, and clarifying the synergistic role of IPC
Developing effective teams and imbedding multi-disciplinary collaboration
Developing synergistic goals and strategies
Developing robust data-driven institutional action plans
Defining optimal evidence-based strategies for cooperative management of patients
Defining optimal outcome metrics of these combined efforts
Tailoring educational strategies to suit all disciplines simultaneously
Utilizing advanced IT tools to support ASP and IPC monitoring and implementation. e.g., multi-dimensional dashboards
Providing system-wide performance and outcome feedback utilizing IT tools
Translation of generated outcomes via enhanced communication strategies to sustain awareness and engagement
Studying the benefits and costs of different combined interventional strategies to provide scientific data that support allocation of increased resources
Establishing universal quantitative end points for IPC and ASP control efforts for AMR organisms

IPC: infection prevention control, IT: information technology, AMR: antimicrobial resistant, and ASP: antibiotic stewardship programme.

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
