# Peer review of "Antimicrobial Stewardship: Leveraging the “Butterfly Effect” of Hand Hygiene"

_antibiotics, 2022, doi:10.3390/antibiotics11101348_

Round 1

Reviewer 1 Report

This paper is more in the nature of a commentary and opinion piece, with joint authorship from the laboratory and the clinical perspective. It presents a strong argument, but could be more clearly structured to assist the reader.

In particular, more explanation in the introduction of the paper on the purpose and rationale of the paper would assist the reader in understanding the ‘scope’ of the paper. Currently the authors attempt to cover both high income and low income settings, with the argument that the recommended approach is applicable in both settings.  While the resource setting is mentioned in some of the cited papers, it is not always mentioned. Greater focus on the argument for the combined approach in low resource settings would strengthen the paper.

The evidence cited also suggests that the focus is on hospital settings, rather than community settings, and potentially particularly on intensive care. This could also be clarified and justified in the introduction.

Comments on specific sections

Abstract: Summarises the argument of the paper, but does not indicate the contribution of the paper / its purpose or its scope.

Introduction

Line 50-52 needs more explanation eg ‘actionable antimicrobial stewardship determinants’ – what does this mean ?

Lines 53-60 focus appears to be on hospital infections, rather than use of antibiotics in the community – clarify focus in introduction, as numbers referred to in para 1 presumably include both community and hospital settings

No statement of the purpose of the paper or its scope, particularly in terms of settings (hospital / community; low income /high income). There is also a need for clarification on the method /basis for the opinions expressed  in the paper, in particular, the use of literature - how were the papers identified / selected ? While this is an ‘opinion’ paper some comment on the availability and selection of literature used would assist the reader.

2. Butterfly effect

In this section, it is not always clear what context is being discussed, in particular, whether this is high resource or low resource settings; in hospital or community settings.

Line 94 ff Potential for contextual dependence – eg is this data focused on high resource or low resource settings ?

Line 117 ff suggests a high resource setting, confirmed in lines 130-131. However, may be clearer for reader to clarify setting at the initial description of the study

Useful to provide a low resource setting example

Line 141 setting not described.

Line 149 ff This appears to be a recommendation that is focused on compliance in regards AMS and IPC, and would be better placed in a section on recommendations eg after section 3

3. Coordinating and integrating IPC and AMS

Line 164-167 More information on the specific challenges in low income countries would support this recommendation

Line 171ff Are these recommendations from the paper referenced (22) or are these developed by the authors ?

Line 182 ff and Table 1 – What is the basis for these recommendations ? Are they based on the literature / references, or developed by the authors ?

In particular, line 185, recommends a more focused approach, for a specific patient /care setting and specific pathogens. What is the rationale for this recommendation ?  What pathogens or settings would the authors suggest might be suitable ? Would this approach be suitable for both high income and low income settings ?

Reviewer 2 Report

Thank you for this article. In my opinion it is higly important to underline that close collaboration between IPC and ASP leadership is needed to ensure the effectiveness of ASP.The manuscript is well written and simple to understand for the readers.

Some minor suggestions:

-line 54 and 55 : it is SARS-COv-2  - it should be SARS-CoV-2

-in references line 255 - there is no reference no 20 - please check it.

Reviewer 3 Report

1. The authors sometimes use ASP to refer to antimicrobial stewardship program, and then call it ASM program in the next paragraph. Probably better to be consistent. 

2. The paper could also be improved if the authors included more details about how COVID impact infection prevention control in good ways and bad ways.

Round 2

Reviewer 1 Report

While the authors have not provided an explanation for their responses, they have addressed the majority of what are essentially suggestions or recommendations in the review of the earlier version. In particular there is greater clarity on the context of the perspective in relation to Covid 19 experience, the key message of the paper (synergy between ASP and IPC), and more reference to the relevance in both high resource and low resource settings. The revisions also clarify the focus on hospital settings. Overall, the revised version addresses the key issues in the review of the original version . However, the current version has edits in track changes, while a clean version will be required for publication. 

Responses to comments on the first version of the paper (in italics)

Abstract: Summarises the argument of the paper, but does not indicate the contribution of the paper / its purpose or its scope - addressed

Introduction

Line 50-52 needs more explanation eg ‘actionable antimicrobial stewardship determinants’ – what does this mean ? - addressed

Lines 53-60 focus appears to be on hospital infections, rather than use of antibiotics in the community – clarify focus in introduction, as numbers referred to in para 1 presumably include both community and hospital settings

Revised additional paragraph clarifies focus on hospital settings and links the paper better to the impacts of the covid 19 pandemic

No statement of the purpose of the paper or its scope, particularly in terms of settings (hospital / community; low income /high income). There is also a need for clarification on the method /basis for the opinions expressed  in the paper, in particular, the use of literature - how were the papers identified / selected ? While this is an ‘opinion’ paper some comment on the availability and selection of literature used would assist the reader.

The additional sentence at the end of this section goes some way to clarifying the purpose of the paper.

2. Butterfly effect

In this section, it is not always clear what context is being discussed, in particular, whether this is high resource or low resource settings; in hospital or community settings.

The revised version provides a clearer statement of context and the rationale for the perspective focus on combining IPC and ASP. The use of the ASP abbreviation is clearer than the previous AMS.

Line 94 ff Potential for contextual dependence – eg is this data focused on high resource or low resource settings ?  - clarified

Line 117 ff suggests a high resource setting, confirmed in lines 130-131. However, may be clearer for reader to clarify setting at the initial description of the study – not addressed

Useful to provide a low resource setting example – not provided, may not be available. However addressed to some extent by comments in lines 178-179 revised version

Line 141 setting not described.- not addressed

Line 149 ff This appears to be a recommendation that is focused on compliance in regards AMS and IPC, and would be better placed in a section on recommendations eg after section 3 – not addressed

3. Coordinating and integrating IPC and AMS

Line 164-167 More information on the specific challenges in low income countries would support this recommendation – specific recommendation provided

Line 171ff Are these recommendations from the paper referenced (22) or are these developed by the authors ? – not addressed, but presumably from authors; comment on setting added to cover both HIC and LMIC.

Line 182 ff and Table 1 – What is the basis for these recommendations ? Are they based on the literature / references, or developed by the authors ? – not addressed but presumably authors

In particular, line 185, recommends a more focused approach, for a specific patient /care setting and specific pathogens. What is the rationale for this recommendation ?  What pathogens or settings would the authors suggest might be suitable ? Would this approach be suitable for both high income and low income settings ? – revised to include reference to resource setting.
